# Effect of Self-Chosen Music in Alleviating the Burden on Family Caregivers of Patients with Advanced Cancer: A Randomised Controlled Trial

**DOI:** 10.3390/ijerph20054662

**Published:** 2023-03-06

**Authors:** Inmaculada Valero-Cantero, Cristina Casals, Milagrosa Espinar-Toledo, Francisco Javier Barón-López, Francisco Javier Martínez-Valero, Nuria García-Agua Soler, María Ángeles Vázquez-Sánchez

**Affiliations:** 1Puerta Blanca Clinical Management Unit, Malaga-Guadalhorce Health District, 29004 Malaga, Spain; 2ExPhy Research Group, Department of Physical Education, University of Cadiz, Puerto Real, 11519 Cadiz, Spain; 3Biomedical Research and Innovation Institute of Cádiz (INiBICA) Research Unit, Puerta del Mar University Hospital, 11009 Cadiz, Spain; 4Rincon de la Victoria Clinical Management Unit, Malaga-Guadalhorce Health District, 29004 Malaga, Spain; 5Department of Preventive Medicine, Public Health and Science History, University of Malaga, 29071 Malaga, Spain; 6Malaga Biomedical Research Institute (IBIMA), 29590 Malaga, Spain; 7Midlothian Foot Care, Dalkeith and National Health Service, Dalkeith EH22 1DU, UK; 8Department of Pharmacology, Faculty of Medicine, University of Malaga, 29010 Malaga, Spain; 9Department of Nursing, Faculty of Health Sciences, University of Malaga, 29071 Malaga, Spain; 10PASOS Research Group, UMA REDIAS Network of Law and Artificial Intelligence Applied to Health and Biotechnology, University of Malaga, 29071 Malaga, Spain

**Keywords:** informal caregivers, caregiving, caregiver strain index, complementary medicine, alternative medicine, music intervention, music medicine, palliative care, oncology, home care

## Abstract

The experience of caregiver burden among family members of patients with advanced cancer is a common problem. The aim of this study was to determine whether the burden may be alleviated by means of a therapeutic approach based on self-chosen music. This randomised controlled trial (ClinicalTrials.gov, NCT04052074. Registered 9 August 2019) included 82 family caregivers of patients receiving home palliative care for advanced cancer. The intervention group (*n* = 41) listened to pre-recorded, self-chosen music for 30 min/day for seven consecutive days, while the control group (*n* = 41) listened to a recording of basic therapeutic education at the same frequency. The degree of burden was assessed by the Caregiver Strain Index (CSI), calculated before and after the seven-day intervention. According to this measure, caregiver burden fell significantly in the intervention group (CSI change: −0.56, SD 2.16) but increased in the control group (CSI change: +0.68, SD 1.47), with a significant group x moment interaction F(1, 80) = 9.30, *p* = 0.003, η^2^_p_ = 0.11. These results suggest that, in the short term at least, the use of therapy based on self-chosen music alleviates the burden on family caregivers of palliative cancer patients. Moreover, this therapy is easy to administer at home and does not present any problems in practice.

## 1. Introduction

Cancer is a major cause of death worldwide. Cancer patients survive longer thanks to the development of modern medicine, especially thanks to the improvements in cancer diagnosis, treatment, and palliative care. Thus, patients with advanced cancer often need palliative care, and the length of the caregiving period has extended. In addition, the symptoms of advanced cancer usually worsen as the disease progresses [1,2], impacting not only on the patients themselves but also their family caregivers [3]. In home palliative care, family caregivers are usually relatives of the patient, most commonly a spouse or daughter, although other family members or friends sometimes perform this role, for which no financial compensation is obtained [3]. A family caregiver has been defined as “any relative, partner, friend or neighbour who has a significant personal relationship with, and provides a broad range of assistance for, an older person or an adult with a chronic or disabling condition” [4].

Despite the benefits of home care in terms of cost-saving and convenience, family caregivers may be affected by numerous problems, including anxiety, depression, fatigue, sleep problems, and reduced quality of life [5,6,7,8,9]. It has been previously confirmed that family caregivers for patients with advanced cancer suffer negative consequences from the presence and intensity of the patients' symptoms [3]. A problem to which family caregivers of patients with advanced cancer are especially susceptible is that of burden, which has been defined as the “negative objective and subjective results stemming from the care taken by the caregiver such as physical, psychological, social and economic problems” [10]. Therefore, attention to caregiving issues is important in understanding what can be done to minimize the burdens of caregiving.

Family caregivers of cancer patients receiving home palliative care usually spend a large part of their day providing accompaniment, care, and attention for long periods. The effects of burden are often heightened during the caregiving process [10], as the duration and degree of daily attention required become greater. This ever-increasing load can provoke or aggravate anxiety [11], depression [9], sleep disturbance [7], addiction to alcohol or drugs [12], and decrease the caregiver’s quality of life [13]. Cancer care is becoming increasingly complex. These mental health issues, or the application of maladaptive strategies, can compromise the quality of care both for the caregiver and for the patient.

It is well recognized that caregiving can adversely affect psychological well-being as well as increase caregiver burden. Increased burden and psychological morbidity have been documented. In view of these considerations, interventions are needed to alleviate the burden experienced by caregivers of patients with advanced cancer [14]. According to randomised clinical trials conducted in this area, psychological improvements have been obtained by various types of interventions, such as psychoeducation, skills training, and therapeutic advice [14,15,16,17]. However, in terms of physical and social health outcomes, the benefits observed cannot be directly related to these interventions, and further studies, based on a rigorous methodology, are necessary to clarify the question [14,15,16,17].

In this respect, complementary therapies, i.e., practices that are not generally considered part of regular treatment in medicine, can be of benefit to the caregiver [18]. Relaxation is a well-established approach for managing stress. Thus, the present study focuses on the use of complementary music medicine, which provokes diverse emotions depending on the characteristics of the music being listened to and has been increasingly recommended to promote relaxation and manage stress [19,20]. In previous research, it has been proposed that eight different mechanisms are involved in the evocation of musical emotions, namely brain stem reflexes, rhythmic entrainment, evaluative conditioning, emotional contagion, visual imagery, episodic memory, musical expectancy, and aesthetic judgment [21].

In the view that this approach might alleviate burden by enhancing the caregiver’s emotional health, more studies focused on the use of music are needed to ensure the well-being of caregivers of cancer patients. Music is being used as a more prominent treatment in the clinical field, reducing burden and improving mood states. Interventions in music medicine have been defined as those sessions in which health professionals offer pre-recorded music for passive listening [22]. Therefore, the caregivers themselves can determine the type of music to be provided, thus maximising the well-being produced [23].

The use of music intervention is still relatively novel, and there is a need to investigate its potential therapeutic use. To our knowledge, few clinical trials have been conducted to analyse the effect of music complementary therapy on the well-being of caregivers of patients with advanced cancer, and the published results are inconsistent [24,25]. If healthcare professionals better understand and support caregivers, their health and the care they provide to patients will be enhanced. Thus, the aim of the present study was to determine, via a randomised controlled trial, whether listening to self-chosen music reduces the burden experienced by family caregivers of palliative cancer patients.

## 2. Materials and Methods

### 2.1. Participants

A total of 82 caregivers participated in the present study: 41 in the intervention group and 41 in the control group. The criteria for inclusion were that the participant should be a family caregiver of a patient with advanced cancer in home palliative care, over 18 years of age, and not have a severe hearing deficiency. The exclusion criteria were that the participant was not acting as a caregiver during the entire seven days of the study period. The caregivers were previously informed about the research project with a full explanation of the risks and benefits of the procedure, both orally and in writing, and gave written consent prior to study participation. Hence, caregivers voluntarily participated in this study, and they were given the option and right to withdraw from it without consequence.

Following previous research describing a music therapy intervention for caregivers [26], and using the Epidat 4.0 software (Xunta de Galicia, Santiago de Compostela, Spain), it was estimated that 34 participants should be included in each group (intervention group and control group), for an alpha error of 0.05 and an estimation error of 0.5. This sample size was then increased by 20% in order to compensate for possible dropouts, thus producing a final sample of 82 caregivers, 41 in each group.

The participants were recruited by reference to the lists of patients included in the Palliative Care Assistance Process in the DIRAYA Digital Health Record maintained at the six primary care centres within the Malaga-Guadalhorce Health District (Andalusia, Spain). Family nurses then telephoned potential participants using convenience sampling.

After the caregivers had been recruited, allocation to the groups was randomised by one of the researchers using 82 opaque envelopes. In 41 of these, the participant was assigned to the control group, and in the other 41 envelopes, the participant was assigned to the intervention group. The envelopes were closed and sealed, and then shuffled and numbered from 1 to 82. The investigator in charge then assigned an envelope to each caregiver, in chronological order, opening the envelope and associating the caregiver with the group specified in the envelope. Neither the study evaluators nor the caregivers participating in the study knew which group they belonged to.

The present study was carried out from July 2020 to September 2021. No participants were lost to follow-up during the study. The flow chart for the study recruitment and procedure is presented in Figure 1.

### 2.2. Design

The study is a randomised, double-blind, multicentre, controlled clinical trial of family caregivers of patients with advanced cancer receiving home palliative care, performed at six primary care centres within the Malaga-Guadalhorce Health District, Andalusia, Spain.

The present study was approved by the Research Ethics Committee of the Province of Malaga on 28 March 2019 (project code: AP-0157-2018) and was conducted in accordance with the Helsinki Declaration of the World Medical Association. The study was registered with ClinicalTrials.gov, code number NCT04052074, on 9 August 2019, as a form of public accountability. All data used in this project fell within the scope of data protection regulations.

### 2.3. Measures

The dependent measure considered in the present study was the burden experienced by the caregiver, as assessed by the Case Management Nurse from the corresponding health centre, during a home visit using the Caregiver Strain Index (CSI). This outcome was assessed by using a CSI questionnaire, which has proven to be reliable and is validated for use with a population of Spanish caregivers [27,28]. The CSI is designed to determine the burden experienced by family and informal caregivers of dependent persons, regardless of the health problem that caused the dependency. The questionnaire consists of 13 true-false propositions, to which each positive response is awarded one point. The total score possible, thus, ranges from 0 to 13 points. The higher the score obtained in the CSI questionnaire, the higher the level of burden presented by the caregiver. A score of seven or more indicates a significant degree of burden.

In addition to this, the sociodemographic characteristics of the participants were registered. Sociodemographic variables included age, sex, marital status, education, paid employment, relationship with the patient, duration of care, and receiving or not receiving assistance for caregiving. These variables were indicated by the caregiver in a personal interview with a nurse prior to the study.

### 2.4. Procedure

The independent variable considered was that of listening to self-chosen music as a complementary intervention. Prior to the intervention, all caregivers (both in the control group and the intervention group) received conventional health care under the same conditions, including basic therapeutic education for palliative care, which is part of the regular health care service provided by nurses at the respective health centres.

The caregivers in the intervention group also received a music-based complementary intervention, listening to pre-recorded music on an mp3 device or mobile phone. To ensure that the intervention enhanced the participants’ well-being, the music selection was chosen by each individual, who provided a list of musical items they liked. This benefit was later confirmed in a personal interview with a nurse, who prepared a playlist accordingly. The personal interview mentioned was the one in which sociodemographic characteristics were also registered. The intervention consisted of a 30 min music session, received daily for seven consecutive days. The caregiver was instructed not to perform any other activity during the session.

The participants in the control group received a recorded repetition of the basic therapeutic training education given (also delivered via an mp3 device or mobile phone), thus masking their membership in the control group both to other participants and to the evaluators. This activity was also performed in 30 min sessions received daily for seven consecutive days, and the caregivers were also instructed not to perform any other activity during the session.

Both groups were advised to listen to the recordings in the morning, when they could be alone. No healthcare professional was present during the listening. The types of music chosen by the intervention group and details of the basic therapeutic education messages given to the control group are described in Table 1.

The CSI data were obtained before and after the intervention for both groups.

### 2.5. Statistical Analyses

Bivariate analyses were performed to detect any demographic or other baseline differences between the control and intervention groups, using the non-parametric Mann–Whitney test, the Chi-square test, or Fisher’s exact test. The CSI differences were analysed by applying a 2 × 2 mixed factorial analysis of variance (ANOVA), with the group (the intervention group vs. the control group) as the inter-subject variable and the moment (pre-intervention vs. post-intervention) as the intra-subject variable. In the Bonferroni post-hoc comparisons performed, a *p* value of <0.05 was considered statistically significant. All analyses were performed with SPSS 23.0 statistical software (SPSS Inc., Chicago, IL, USA). Epidat software version 4.0 (Xunta de Galicia, Santiago de Compostela, Spain) was used to calculate the sample size.

## 3. Results

The bivariate analyses revealed no significant differences in sociodemographic or other baseline data between the control and intervention groups (Table 2).

There was a significant group x moment interaction (F (1, 80) = 9.30, *p* = 0.003, η^2^_p_ = 0.11) in caregiver burden of family caregivers of palliative cancer patients. A significant decrease in caregiver burden was observed in the intervention group but not in the control group, showing statistically significant differences. The CSI values obtained before and after the seven-day intervention are shown in Table 3.

During the intervention, none of the participants presented any adverse events.

## 4. Discussion

Caregivers are at high risk for stress, poor quality of life, and burnout. To address these concerns, increasing attention has been paid to using music in clinical settings such as cancer centres, hospices, and palliative programmes [29]. The results obtained in this clinical trial show that complementary music medicine for family caregivers of patients with advanced cancer, using self-chosen music, may reduce the burden compared to usual care. However, it should be noted that the intervention considered was very brief in relation to the entire duration of the treatment, and the findings should be interpreted with caution.

According to our results, an earlier clinical trial reported that listening to pre-recorded music alleviated anxiety symptoms and depression in cancer patient caregivers [25], conditions that are associated with burden [9,11]. Our own study detected a short-term burden alleviation in the intervention group. The findings add to the growing body of research linking music intervention to psychological well-being. However, the burden on family caregivers of patients with advanced cancer usually increases as the process advances [10], and therefore further investigation featuring a longer follow-up is needed.

Another important consideration is that the long-term alleviation of caregiver burden could also enhance the quality of palliative care provided [30]. Music-related activities may induce relaxation, reduce caregiver stress [31], facilitate the expression of thoughts and feelings, and enable the recovery of recollections [32]. The mechanism underlying these effects is still not clearly understood, but in contrast to the impact of a stressful situation, relaxing music is known to modulate the autonomous nervous system [33] and lower plasma cortisol levels [34], among other physiological responses associated with stress reduction and enhanced well-being. Nevertheless, researchers have yet to establish whether any specific type of music is more appropriate in this respect [25].

In the present study, among the twelve musical genres offered to the caregivers in the intervention group, those most frequently chosen were flamenco, Latin pop, and lounge music. Some participants, therefore, opted for more relaxing music but the majority were not guided by that consideration, but rather by their preferred type of music. In further study, this question should be studied in greater depth to determine whether the musical genre chosen influences the degree of alleviation obtained. In addition, further research is required to determine whether there is any continuous effect on family caregivers of patients with advanced cancer.

Among the caregivers in our sample, 57% presented a high level of burden, which is in line with previous research findings [35]. Most were female, had no external employment, and did not receive assistance for caregiving, which also corroborates previous reports [35] and reflects the representativeness of our sample. Alternatively, a study conducted with caregivers of patients with advanced cancer or chronic disease [30] reported higher levels of burden than in our case (mean CSI score: 8.5 vs. 6.8 in our study population). In accordance with these observations, it would be useful to test the effectiveness of self-chosen music-based therapy for caregivers in this situation (i.e., one that presents a higher level of burden).

In summary, caregiver burden is frequently experienced by those caring for family members with advanced cancer. In this study, we report the effectiveness of a short-term complementary intervention that is easily administered in a clinical setting and benefits the well-being of these family caregivers, though some limitations must be addressed. Such an intervention can be indicated by nurses providing home attention to enhance the care and health status of caregivers, which would also benefit the patients.

## 5. Limitations

The present study is subject to some limitations that must be acknowledged. Firstly, the participants were recruited at urban public health centres, and their characteristics might differ from those of caregivers attending private health centres or residents of other geographical areas. Consequently, the generalisation of our results may be limited. Moreover, as there were more caregivers with higher CSI scores (>7) in the intervention group (61%) than in the control group (54%) at baseline, the findings should be interpreted with caution. However, these differences were slight and not statistically significant.

The internal validity of the trial could be compromised if the participants in the control group guessed their placebo condition. In the event of failed blinding, a serious limitation to the internal validity of the trial would be presented due to the fact that patients' perception of treatment are related to their treatment responses in some situations. In order to minimise the Hawthorne effect, all participants were personally interviewed by a case management nurse to obtain the necessary sociodemographic information and explain the procedures involved (including instructions for both groups regarding the use of the audio listening device). In addition, the members of the intervention group were asked about their preferred style of music. This question was not asked of the control group in order to prevent them from realising they were not receiving the intervention. Nevertheless, this difference in the questions asked of the two groups and the consequent Hawthorne effect may be considered a limitation.

Despite these limitations, the rigorous nature of the clinical trial conducted supports the validity of the conclusions drawn. Future research should elucidate the long-term effect of music listening on reducing caretaker burden, as well as its relation to other health outcomes such as quality of life, anxiety, and depression.

## 6. Conclusions

This study shows that daily listening to self-selected music for thirty minutes over seven consecutive days can reduce the subjective burden and thus enhance the well-being of family caregivers of palliative cancer patients. Self-chosen music listening may serve as an effective coping strategy for family caregivers; this intervention has no side effects and can be applied in conjunction with usual short-term clinical practice. The study findings underscore the need for complementary, cost-effective, and accessible strategies to alleviate the caregiver burden of patients with at-home palliative care.

## Figures and Tables

**Figure 1 ijerph-20-04662-f001:**
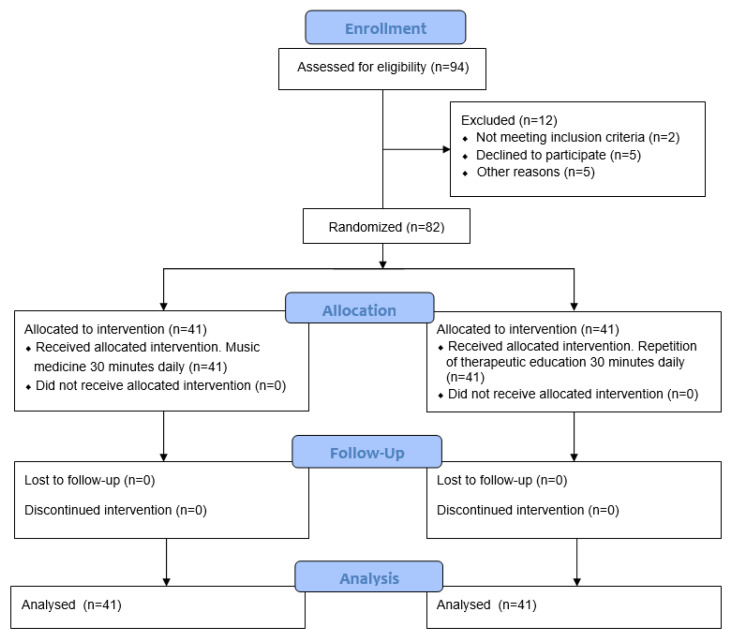
Flow chart for recruitment and procedures of the study.

**Table 1 ijerph-20-04662-t001:** Audio content for the intervention and control groups of family caregivers.

Intervention Group (*n* = 41)		Control Group (*n* = 41)	
Type of Music Chosen	N	Repetition of Basic Therapeutic Education	N
Flamenco	7	Food and hydration	41
Latin pop	7	Exercise and leisure	41
Lounge	7	Medication	41
Pop rock	4	Effective communication	41
Romantic ballad	4	Skin care	41
Celtic	2	Prevention of constipation	41
Flamenco pop	2		
Flamenco/Spanish rock	2		
Opera	2		
Classical Indian	1		
American rock	1		

**Table 2 ijerph-20-04662-t002:** Bivariate analysis of socio-demographic data for caregivers of palliative cancer patients.

Characteristics	Total (*n* = 82)	Intervention Group (*n* = 41)	Control Group (*n* = 41)	*p* Value
Age				
Years	62.71 ± 12.5	63.12 ± 12.8	62.31 ± 12.5	0.626
Sex				
Female	72 (87.8%)	34 (82.9%)	38 (92.7%)	0.177 ^¶¶^
Male	10 (12.2%)	7 (17.1%)	3 (7.3%)	
Marital status				
Married	51 (62.2%)	27 (65.9%)	24 (58.5%)	0.913 ^¶¶¶^
Single	15 (18.3%)	7 (17.1%)	8 (19.5%)	
Divorced	14 (17.1%)	6 (14.6%)	8 (19.5%)	
Widowed	2 (2.4%)	1 (2.4%)	1 (2.4%)	
Employment status				
No external work	39 (47.6%)	20 (48.8%)	19 (46.3%)	0.965 ^¶¶¶^
Working outside the home	23 (28.0%)	11 (26.8%)	12 (29.3%)	
Unemployed	9 (11.0%)	4 (9.8%)	5 (12.2%)	
Retired	11 (13.4%)	6 (14.6%)	5 (12.2%)	
Duration of care				
Months	18.27 ± 28.9	17.43 ± 29.0	19.12 ± 29.1	0.809
Receiving assistance for caregiving				
Yes	27 (65.9%)	13 (31.7%)	14 (34.1%)	0.814 ^¶¶^
No	55 (67.1%)	28 (68.3%)	27 (65.9%)	
Caregiver Strain Index				
≥7	47 (57.3%)	25 (61.0%)	22 (53.7%)	0.503 ^¶¶^
<7	35 (42.7%)	16 (39.0%)	19 (46.3%)	

The results are shown as mean ± standard deviation and as number of participants (percentage) for frequencies. Mann–Whitney U test was applied unless otherwise stated. ^¶¶^ means Chi-square test. ^¶¶¶^ means Fisher’s exact test.

**Table 3 ijerph-20-04662-t003:** Caregiver Strain Index scores of family caregivers before and after the intervention.

	Intervention Group (*n* = 41)	Control Group (*n* = 41)	*p* Value
Baseline scores	6.80 ± 3.23	6.83 ± 2.81	0.003
Post-intervention scores	6.24 ± 3.30	7.51 ± 2.80 *	

The results are shown as mean ± standard deviation. * means *p* < 0.05 between the intervention and control groups according to Bonferroni post-hoc comparisons.

## Data Availability

Study data are available on reasonable request to the corresponding author.

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
