# Peer review of "Effect of Self-Chosen Music in Alleviating the Burden on Family Caregivers of Patients with Advanced Cancer: A Randomised Controlled Trial"

_ijerph, 2023, doi:10.3390/ijerph20054662_

Round 1
Reviewer 1 Report
The authors report an RCT on music listening as a strategy to reduce family caregiver burden for individuals who care for family members with advanced cancer.
I suggest to present the design of the study as a 2x2 ANOVA design, with time and group as independent, and CSI as dependent measures.
Abstract
„but this hypothesis has not been…” Please do not speculate about what was not done, rather stick to what you did. “Here we tested the assumption that…”
You say it is an RCT, so no need to reiterate that individuals were “randomized”
Please offer more detail about how music was selected: this is a bit obscure and not well substantiated throughout the manuscript.
Report significant interaction (if observed) following revised statistical procedures. Also provide posthoc mean comparison tests to substantiate possible interactions (this is my guess).
Introduction
Better define “family caregiver” and “cargiver burden” early on to use these terms consistently onwards.
l.50ff: “These aspects…” no “These mental health issues, or maladaptive strategies can compromise the quality of care on both sides, carer and patient.” Something like that.
l.56: Please use refs after each line, not just at the end of the para. The need for efficient and cost-effective strategies, that are easy to implement, accessible to most carers, and free of severe side effects is certainly described in the literature. So please use refs after such lines.
l.72. Please name authors of relevant studies, briefly describe their strategies and outcomes. In brief, add more detail.
L. 80: “The absence of any prior study… “ No, that is poor phrasing. Please include a subheading “The present study” Therein, please describe precisely the aims, the research questions, and hypotheses to turn back to in the Discussion part later.
Write about participant-selected and experimenter-selected music as relevant strategies and back up your decision to use participant-selected music based on literature, rather than pure intuition.
Matierals and Methods
We know already about the RCT, because the rationale will be described under “The present study” above.
Please omit any superfluous or redundant information. Consult APA to learn about the subheadings of a Methods section
Better start with
“Participants”
Then “Design and Procedures”
Within this section, describe
- Dependent measures (provide example items and psychometric properties of the CSI scale; especially re-test reliability as you use it at two time points)
- Independent measures
- Selection procedures for the music and verbal materials (in detail).
- Implementation of protocol. Who listened to what (context, time of day, alone, in presence of others etc., headphones suggest individual listening, but reveal nothing about context. Were time of day controlled for intervention and comparison group?-
Results
Figure one is clearly not a result, but should go above under “Methods”
Table 2 is more or less fine, as it describes the demographics
Table 3 should depict descriptive stats for outcome variables across groups and time points. Consult APA for an appropriate way to construct the Table.
Table 4 should hold the inferential stats using an 2x2 ANOVA to restrict inflation of Type I errors
Figure 2 should give a bar chart with the presumed interaction effect (please use bars depicting SD or SE), if you find one.
Discussion
The aim should be described at the end of the intro, (The present study), along with RQs and hypotheses. Here you address the latter to, not the aim.
L.185: “is a finding of high importance.” – Avoid such phrasing that does not merit any content.
The last paragraph offers “Limitations”, please use a subheading because there is much more.
Seven days within an average of 12-18 months of palliative care is not very long. So the authors are not in a position for strong claims. It may be no more than a temporary relief, that washes off quickly as the disease proceeds, who knows.
The authors do not report any interviewing about how music and verbal materials were rated by the participants. That could be key to understanding why the control did not work so well. The rationale for selecting verbal materials is not clear to me at all. The authors should better explain that.
The authors do not offer any model, mechanism, or insight into how the effect was mediated. Was it an improvement of mood? A reduction of stress? Both, or neither of that, just some prescription that did the trick (Hawthorne effect); as nothing is known as to how the comparison condition was received, it is unclear what the music did and what it per se achieved. How about a good audio book as comparison?
Conclusions
“no negative consequences”… what does that mean?
Author Response
Response to Reviewer 1 Comments
Point 1: The authors report an RCT on music listening as a strategy to reduce family caregiver burden for individuals who care for family members with advanced cancer. I suggest to present the design of the study as a 2x2 ANOVA design, with time and group as independent, and CSI as dependent measures.
Response 1: There was a significant interaction group x moment F (1, 80) = 9.30, p = 0.003, η2p = 0.11.
Point 2: Abstract „but this hypothesis has not been…” Please do not speculate about what was not done, rather stick to what you did. “Here we tested the assumption that…”
Response 2: Corrected after suggestion.
Point 3: You say it is an RCT, so no need to reiterate that individuals were “randomized”
Response 3: We agree. It has been removed.
Point 4: Please offer more detail about how music was selected: this is a bit obscure and not well substantiated throughout the manuscript.
Response 4: We agree with the reviewer. In fact, other reviewer also has recommended to better detail this process in the Method. Thus, we have included this information, however, it is complicated for us to include it in the Abstract due to the word limit. We have specified the term “self-chosen music” a deeply detailed in the Method. Thank you very much.
Point 5: Report significant interaction (if observed) following revised statistical procedures.
Response 5: The factorial analysis of variance has been included according to your suggestion.
Point 6: Also provide posthoc mean comparison tests to substantiate possible interactions (this is my guess).
Response 6: Yes, Bonferroni post hoc comparisons have been applied. Thank you. However, we have the aforementioned concern regarding the number of words allowed in the abstract. The results section has been modified.
Point 7: Introduction. Better define “family caregiver” and “cargiver burden” early on to use these terms consistently onwards.
Response 7: The caregiver burden was defined in the first draft of the manuscript (l.47) appearing in the introduction and defined as the “negative objective and subjective results stemming from the care taken by the caregiver such as physical, psychological, social, and economic problems” [9]. We have included the family caregiver definition which was not previously included, it is defined as follows: “A family caregiver has been defined as “any relative, partner, friend or neighbour who has a significant personal relationship with, and provides a broad range of assistance for, an older person or an adult with a chronic or disabling condition”.
Point 8: l.50ff: “These aspects…” no “These mental health issues, or maladaptive strategies can compromise the quality of care on both sides, carer and patient.” Something like that.
Response 8: Corrected after suggestion.
Point 9: l.56: Please use refs after each line, not just at the end of the para. The need for efficient and cost-effective strategies, that are easy to implement, accessible to most carers, and free of severe side effects is certainly described in the literature. So please use refs after such lines.
Response 9: Corrected after suggestion.
Point 10: l.72. Please name authors of relevant studies, briefly describe their strategies and outcomes. In brief, add more detail.
Response 10: The information is presented in order to lead the reader to our aim. However, the discussion section has been also expanded.
In the study developed by Choi the music intervention consisted of four sessions: two 30-minute sessions a week for 2 weeks, the results of this study showed a significant decrease in anxiety and fatigue and an increase in quality of life, although with no significant difference between groups. In another study conducted by Lai et al., it was compared active and passive music listening methods on a single 30-minute session, they found a decrease in depression, anxiety, and blood volume pulse amplitude in both groups.
Point 11: L. 80: “The absence of any prior study… “ No, that is poor phrasing. Please include a subheading “The present study” Therein, please describe precisely the aims, the research questions, and hypotheses to turn back to in the Discussion part later.
Response 11: It has been rewritten according to the reviewer’s suggestion. Thank you for your comments.
Point 12: Write about participant-selected and experimenter-selected music as relevant strategies and back up your decision to use participant-selected music based on literature, rather than pure intuition.
Response 12: We do not aim to compare different music interventions, thus, we are not comparing or describing different models.
Point 13: Matierals and Methods. We know already about the RCT, because the rationale will be described under “The present study” above.
Response 13: Corrected after suggestion.
Point 14: Please omit any superfluous or redundant information. Consult APA to learn about the subheadings of a Methods section
Response 14: We have reviewed the method section. Thank you for your suggestions.
Point 15: Better start with “Participants”. Then “Design and Procedures” Within this section, describe
- Dependent measures (provide example items and psychometric properties of the CSI scale; especially re-test reliability as you use it at two time points)
- Independent measures
- Selection procedures for the music and verbal materials (in detail).
- Implementation of protocol. Who listened to what (context, time of day, alone, in presence of others etc., headphones suggest individual listening, but reveal nothing about context. Were time of day controlled for intervention and comparison group?-
Response 15: Although we appreciate your comments, we prefer to start with design as is common in several journals also. We have carefully revised this section. We also prefer to present the independent measures (experimental vs. control) before the dependent measures. We have tried to better detail this section. Psychometric properties of the CSI has been included giving this relevant information according to the reviewer’s suggestion in the hope that it is suitable for publication.
Point 16: Results. Figure one is clearly not a result, but should go above under “Methods”
Response 16: Corrected after suggestion.
Point 17: Table 2 is more or less fine, as it describes the demographics
Response 17: Ok, thanks.
Point 18: Table 3 should depict descriptive stats for outcome variables across groups and time points. Consult APA for an appropriate way to construct the Table.
Response 18: We have not included a Table 3. The changes in our outcome variable for each before and after the intervention is detailed in the text.
Point 19: Table 4 should hold the inferential stats using an 2x2 ANOVA to restrict inflation of Type I errors
Response 19: We have not included a Table 4. Factorial ANOVA is detailed in the text.
Point 20: Figure 2 should give a bar chart with the presumed interaction effect (please use bars depicting SD or SE), if you find one.
Response 20: We have not included a Figure 2 in order to avoid duplicated information. Factorial ANOVA is detailed in the text as well as mean and SD.
Point 21: Discussion. The aim should be described at the end of the intro, (The present study), along with RQs and hypotheses. Here you address the latter to, not the aim.
Response 21: Corrected after suggestion.
Point 22: L.185: “is a finding of high importance.” – Avoid such phrasing that does not merit any content.
Response 22: Rewritten after suggestion although we do not understand the concern, we beg to differ.
Point 23: The last paragraph offers “Limitations”, please use a subheading because there is much more.
Response 23: As there would be only a subheading we prefer to avoid it. However, we have expanded the limitations.
Point 24: Seven days within an average of 12-18 months of palliative care is not very long. So the authors are not in a position for strong claims. It may be no more than a temporary relief, that washes off quickly as the disease proceeds, who knows.
Response 24: It has been included. Thank you.
Point 25: The authors do not report any interviewing about how music and verbal materials were rated by the participants. That could be key to understanding why the control did not work so well. The rationale for selecting verbal materials is not clear to me at all. The authors should better explain that.
Response 25: It has been better detailed in methods.
Point 26: The authors do not offer any model, mechanism, or insight into how the effect was mediated. Was it an improvement of mood? A reduction of stress? Both, or neither of that, just some prescription that did the trick (Hawthorne effect); as nothing is known as to how the comparison condition was received, it is unclear what the music did and what it per se achieved. How about a good audio book as comparison?
Response 26: We have tried to discuss these aspects. However, our aim was not to compare different interventions, such as music vs. a good audio book, maybe both interventions are suitable but it is still needed a control group in this example.
Point 27: Conclusions “no negative consequences”… what does that mean?
Response 27: Corrected “without side effects”. Thank you for your time and all the comments in order to improve this manuscript.

Reviewer 2 Report
Efficacy of Pre-Recorded Music in Alleviating Burden for Family Caregivers of Patients with Advanced Cancer: A Randomised Controlled Trial
Thank you for your invitation to review this manuscript of a RCT of a music therapy programme for caregivers of patients with advanced cancer. The intervention group (n=41) listened to pre-recorded music for 30 min/day during 7-consecutive days, while the control group (n=41) listened to a recording of basic therapeutic education. The primary outcome reported here is Caregiver Strain Index (CSI). The authors reported that caregiver burden fell significantly in the intervention group (CSI change: -0.56, SD 2.16), but increased in the control group (CSI change: +0.68, SD 1.47), p <0.001.
I’d like to congratulate the authors for conducting an RCT on this topic. (1) The findings are encouraging, however, overall, I found there is a lack of debates on the current evidence in relation to listening to self-chosen music and its physiological and psychological effects. Perhaps, literature in cancer/palliative care may be limited, but certainly, there is vast research-based evidence in music and emotion regulation. (2) I’ve checked the RCT registration, protocol and reviewed this manuscript using the CONSORT checklist. This manuscript doesn’t report all outcome measures listed in the protocol. I’d like to question the authors about this.
Here are some suggestions to improve the manuscript.
1. Title: Identification as a randomised trial in the title is included. I think that preferred music or self-chosen music is more important here than pre-recorded music.
2. Please ensure that the abstract and the whole manuscript are written in high-standard English.
3. Key words: I’d suggest that listening to music, or music intervention should be included. Complementary therapy; Alternative therapy are very similar, therefore, including one of them would be sufficient.
4. Introduction: Line 77-78 – it is unclear (incorrect?) what the previous study has found (ref #23).
5. Introduction: Line 82-84 – Based on the CONSORT checklist, could the authors include clearer objectives or hypotheses?
6. Materials and Methods: Line 94: it’s important to state whether participants’ consents were received prior to study participation.
7. Materials and Methods: Line 95-96 – sample size calculation. Could the authors please explain why the power calculation is based on satisfaction (in the ‘Happiness’ scale)? Conventionally, the power calculation is based on the primary outcome measure.
8. Materials and Methods: Line 110 & 129 – Double blind? Were ALL caregivers (i.e., both intervention and control groups) OR only the intervention group caregivers asked to provide their preferred music list (presumably prior to study commence)? How was it possible to conceal the control group caregivers from their assigned condition when they were not asked to provide music list and when they listened to educational, spoken words??
9. Materials and Methods: Line 117 & 123 repeated sentences “The music used was individualised, chosen by each participant to ensure that the music enhanced their well-being”.
10. Materials and Methods: Other outcome measures that were stated in the trial protocol, such as the Family Quality of Life (FQOL) scale, Client Satisfaction Questionnaire (CSQ-8) were not reported here.
11. Figure 1. I suggest revising the Flow diagram according to CONSORT 2010 Statement: http://www.consort-statement.org/Media/Default/Downloads/CONSORT%202010%20Statement/CONSORT%202010%20Statement%20(BMJ).pdf
12. Discussion: Line 176-186 – I suggest the authors reflect on the findings with more caution, as there were more caregivers with higher Caregiver Strain Index scores (>7) in the intervention group (61%) then in the control (53.7%) at the baseline.
13. Discussion: Line 194 – “complementary music medicine” I’d like to ask the authors what qualifies as complementary music medicine. I don’t think it’s a well-known term. In this trial, there is no mention of how the participants made their own list of music. Was there a therapist/health professional assisting with this? If they had assistance with music therapists or nurses for example, maybe it can be called as ‘music medicine’. But, anyone can make their own list of self-chosen music. Further, everyday, everyone is exposed to music on radio, or shops, gym or even a lot of people already have their personalised lists of music on their phones. How do you define ‘complementary music medicine’ as an intervention in the context of palliative care? Can anyone make their own music list and listen to it, can’t they?
14. Discussion: Line 208-215 - Trial limitations: (a) please could the authors consider other music exposes that trial participants may have experienced? E.g., they may have heard music on radio, at shops, gym, friends’ house etc. How can you be certain that the 30-min listening to music made this difference? Also, how could you prevent the control group participants listening to music outside of the 30-min trial setting? They might listen to music in their car while driving to the trial place?? They go to gym and exercise while listening to music? (b) Why 7 days? Wouldn’t it be important to look at long-term impacts of music listening, given the rationale that caregivers’ burden would increase with time? (c)How about Family Quality of Life (FQOL) scale and Client Satisfaction Questionnaire (CSQ-8)? Were there any other improvements? Perhaps, there was some improvements on the satisfaction questionnaire, which would support the findings of reduced burden?? These weren’t reported here. Why?
15. Discussion: What are the implications for the cancer care and palliative care practice? Could the authors include some recommendations how other palliative care centres could encourage caregivers listen to their preferred music? And how the health professionals can support that? Is there specific music that is more beneficial??
16. Discussion: CONSORT checklist #22 considering other relevant evidence – more relevant evidence should be included in discussion. E.g., a systematic review examined the benefits of creative interventions including music/singing for caregivers. https://www.ncbi.nlm.nih.gov/pmc/articles/PMC7721165/ Compare with group-based active interventions for caregivers, what are the advantages and disadvantages of the listening to music on your own?
17. Conclusions: Line 217-219 – this is an implication for the practice. Please move it up to the Discussion.
18. Conclusions: Please revise the conclusion section with the precise information on the trial aim, findings, limitations, its contribution, and recommendations for future studies.
Author Response
Response to Reviewer 2 Comments
Point 1: Thank you for your invitation to review this manuscript of a RCT of a music therapy programme for caregivers of patients with advanced cancer. The intervention group (n=41) listened to pre-recorded music for 30 min/day during 7-consecutive days, while the control group (n=41) listened to a recording of basic therapeutic education. The primary outcome reported here is Caregiver Strain Index (CSI). The authors reported that caregiver burden fell significantly in the intervention group (CSI change: -0.56, SD 2.16), but increased in the control group (CSI change: +0.68, SD 1.47), p <0.001. I’d like to congratulate the authors for conducting an RCT on this topic. (1) The findings are encouraging, however, overall, I found there is a lack of debates on the current evidence in relation to listening to self-chosen music and its physiological and psychological effects. Perhaps, literature in cancer/palliative care may be limited, but certainly, there is vast research-based evidence in music and emotion regulation.
Response 1: We agree with the reviewer and we have try to better detail it both in the introduction and discussion. All changes have been highlighted in yellow.
Point 2: (2) I’ve checked the RCT registration, protocol and reviewed this manuscript using the CONSORT checklist. This manuscript doesn’t report all outcome measures listed in the protocol. I’d like to question the authors about this.
Response 2: Although we include in the RCT registration all the outcomes, it should be clarified that in the initial proposal that was applied to the funding source (Junta de Andalucía) it is specified that a scientific article will be written for each specific objective proposed. Therefore, the research team of the funded proposal is committed to carrying out the different articles individually as described in the project. Moreover, it is considered that the data presented in the current article have sufficient clinical importance to be published in its current state.
Point 3: Here are some suggestions to improve the manuscript. Title: Identification as a randomised trial in the title is included. I think that preferred music or self-chosen music is more important here than pre-recorded music.
Response 3: We agree with the reviewer. Thus, we have included “self-chosen” in the tittle, also in some key sections such as abstract, aim of the study… Thank you very much.
Point 4: Please ensure that the abstract and the whole manuscript are written in high-standard English.
Response 4: The manuscript have been reviewed by the translator of our University (University of Málaga).
Point 5: Key words: I’d suggest that listening to music, or music intervention should be included. Complementary therapy; Alternative therapy are very similar, therefore, including one of them would be sufficient.
Response 5: Corrected after suggestion.
Point 6: Introduction: Line 77-78 – it is unclear (incorrect?) what the previous study has found (ref #23).
Response 6: It has been rewritten as follows “another study did find significant improvements on anxiety and depression levels after a music intervention in cancer patient caregivers”.
Point 7: Introduction: Line 82-84 – Based on the CONSORT checklist, could the authors include clearer objectives or hypotheses?
Response 7: Corrected after suggestion with “The aim of the study was to determine whether listening to self-chosen music impacts on burden experienced by caregivers of palliative cancer patients”.
Point 8: Materials and Methods: Line 94: it’s important to state whether participants’ consents were received prior to study participation.
Response 8: It has been specified according to the reviewer’s comment. Thank you.
Point 9: Materials and Methods: Line 95-96 – sample size calculation. Could the authors please explain why the power calculation is based on satisfaction (in the ‘Happiness’ scale)? Conventionally, the power calculation is based on the primary outcome measure.
Response 9: We used this outcome when we wrote the proposal in order to obtaining funding and no other similar articles were found. The cited manuscript describes the effect of listening to music on several subjective variables of the caregiver, showing both the mean of the effect and its standard deviation. We selected "Happiness" because it presented a slightly higher standard deviation, which would allow us to obtain a sufficient sample for our investigation. Notwithstanding, this part has been rewritten.
Point 10: Materials and Methods: Line 110 & 129 – Double blind? Were ALL caregivers (i.e., both intervention and control groups) OR only the intervention group caregivers asked to provide their preferred music list (presumably prior to study commence)? How was it possible to conceal the control group caregivers from their assigned condition when they were not asked to provide music list and when they listened to educational, spoken words??
Response 10: In our opinion, if we ask for preferred music but we applied the educational program, this group would know that they are not in the intervention group. All they have an intervention in which they must listen to a record session of similar duration and frequency with the same device.
Point 11: Materials and Methods: Line 117 & 123 repeated sentences “The music used was individualised, chosen by each participant to ensure that the music enhanced their well-being”.
Response 11: Thank you very much. The duplicated paragraph has been removed.
Point 12: Materials and Methods: Other outcome measures that were stated in the trial protocol, such as the Family Quality of Life (FQOL) scale, Client Satisfaction Questionnaire (CSQ-8) were not reported here.
Response 12: Although the trial protocol includes more outcomes, we are presenting each specific aims on specific manuscripts.
Point 13: Figure 1. I suggest revising the Flow diagram according to CONSORT 2010 Statement: http://www.consort-statement.org/Media/Default/Downloads/CONSORT%202010%20Statement/CONSORT%202010%20Statement%20(BMJ).pdf
Response 13: The Flow diagram is presented in the new Figure 1. Thank you.
Point 14: Discussion: Line 176-186 – I suggest the authors reflect on the findings with more caution, as there were more caregivers with higher Caregiver Strain Index scores (>7) in the intervention group (61%) then in the control (53.7%) at the baseline.
Response 14: We agree and we have improved the statistic analyses (also according to another reviewer, the Reviewer 1). The mentioned differences between groups at baseline were not statistically significant. Notwithstanding, we have included this statement in limitations.
Point 15: Discussion: Line 194 – “complementary music medicine” I’d like to ask the authors what qualifies as complementary music medicine. I don’t think it’s a well-known term. In this trial, there is no mention of how the participants made their own list of music. Was there a therapist/health professional assisting with this? If they had assistance with music therapists or nurses for example, maybe it can be called as ‘music medicine’. But, anyone can make their own list of self-chosen music. Further, everyday, everyone is exposed to music on radio, or shops, gym or even a lot of people already have their personalised lists of music on their phones. How do you define ‘complementary music medicine’ as an intervention in the context of palliative care? Can anyone make their own music list and listen to it, can’t they?
Response 15: We have better detailed this information in the Methods section. Moreover, we have remove the term of “complementary music medicine” since we do not think that this is the best definition and we agree with the reviewer’s comments.
Point 16: Discussion: Line 208-215 - Trial limitations: (a) please could the authors consider other music exposes that trial participants may have experienced? E.g., they may have heard music on radio, at shops, gym, friends’ house etc. How can you be certain that the 30-min listening to music made this difference? Also, how could you prevent the control group participants listening to music outside of the 30-min trial setting? They might listen to music in their car while driving to the trial place?? They go to gym and exercise while listening to music?
Response 16: Although we understand this point of view, as we have a randomized control group we can assume that their exposure to “external music” are similar. It is a RCT. Also, we do not want to limit other aspects of their lives, we aimed to investigate whether implement additional 30-min sessions can improve burden, as it occurred, so nurses are encouraged to applied this complementary therapy.
Point 17: Why 7 days? Wouldn’t it be important to look at long-term impacts of music listening, given the rationale that caregivers’ burden would increase with time?
Response 17: Yes, it can be a valid future line of research so we have included it.
Point 18: How about Family Quality of Life (FQOL) scale and Client Satisfaction Questionnaire (CSQ-8)? Were there any other improvements? Perhaps, there was some improvements on the satisfaction questionnaire, which would support the findings of reduced burden?? These weren’t reported here. Why?
Response 18: We have also detailed it in limitations as follows: “Future research should elucidate long-term impacts of music listening on burden, as well as its relation with other health outcomes such as quality of life, anxiety or depression in caregivers of cancer patients”.
Point 19: Discussion: What are the implications for the cancer care and palliative care practice? Could the authors include some recommendations how other palliative care centres could encourage caregivers listen to their preferred music? And how the health professionals can support that? Is there specific music that is more beneficial??
Response 19: The discussion section has been expanded. Thank you.
Point 20: Discussion: CONSORT checklist #22 considering other relevant evidence – more relevant evidence should be included in discussion. E.g., a systematic review examined the benefits of creative interventions including music/singing for caregivers. https://www.ncbi.nlm.nih.gov/pmc/articles/PMC7721165/ Compare with group-based active interventions for caregivers, what are the advantages and disadvantages of the listening to music on your own?
Response 20: This manuscript does not aim to compare different music interventions (creative sessions, active music, etc.) although we have enjoyed the reading of that review. Thank you. The Discussion section has been expanded.
Point 21: Conclusions: Line 217-219 – this is an implication for the practice. Please move it up to the Discussion.
Response 21: Done after suggestion.
Point 22: Conclusions: Please revise the conclusion section with the precise information on the trial aim, findings, limitations, its contribution, and recommendations for future studies.
Response 22: Conclusions have been rewritten in order to better respond to the aim. However, limitations and future studies are presented in the Discussion section.

Reviewer 3 Report
Reviewer Feedback for Efficacy of Pre-Recorded Music in Alleviating Burden for Family Caregivers of Patients with Advanced Cancer: A Randomised Controlled Trial
Thank you for your paper on the important topic of reducing caregiver burden for individuals providing care support to patients with advanced cancer.
Some additional information is recommended as follows:
1. How were the participants recruited? Poster? Email invitation? A referral process?
2. Who prepared the playlists? Did the participants create their own recorded playlist to listen to? Did the participants submit their playlist to the study and have the recorded list prepared for them?
3. More discussion on the role of music:
a. Line 66-68: You state that music provokes diverse emotions. Further discussion on why/how music evokes an emotional response would be helpful.
b. In Discussion Section: more discussion regarding WHY music might have contributed to results.
There is a strong literature base regarding the therapeutic use of music to reduce anxiety or depression. Including a bit more discussion on why music may have resulted in reduced caregiver burden would strengthen conclusions.
A few minor editing items:
Line 40: no comma needed after “death”.
Line 44: no comma needed after “problems”.
Line 66-68: worded awkwardly
Author Response
Response to Reviewer 3 Comments
Point 1: Thank you for your paper on the important topic of reducing caregiver burden for individuals providing care support to patients with advanced cancer. Some additional information is recommended as follows. How were the participants recruited? Poster? Email invitation? A referral process?
Response 1: We agree with the reviewer. This information have been included in the Methods section. All changes have been highlighted in yellow. It has been detailed as follows: “The participants were recruited by reference to the lists of patients included in the Palliative Care Assistance Process in the DIRAYA Digital Health Record maintained at the six primary care centres within the Malaga-Guadalhorce Health District (Spain). Family nurses phoned to the participants using convenience sampling”.
Point 2: Who prepared the playlists? Did the participants create their own recorded playlist to listen to? Did the participants submit their playlist to the study and have the recorded list prepared for them?
Response 2: The music used was individualised, chosen by each participant to ensure that the music enhanced their well-being. We have better detailed this information according to the reviewer’s suggestion as follows: “The participants reported the list of songs that made them feel good, they sent the list to the nurse who checked with the participant that each song made them feel good in a personal interview. Then, the nurse prepared the playlist to each participant”.
Point 3: More discussion on the role of music. Line 66-68: You state that music provokes diverse emotions. Further discussion on why/how music evokes an emotional response would be helpful.
Response 3: We have included the following statement: “It has been previously proposed that eight different mechanisms are involved in the evocation of musical emotions which are brain stem reflexes, rhythmic entrainment, evaluative conditioning, emotional contagion, visual imagery, episodic memory, musical expectancy, and aesthetic judgment”.
Point 4: In Discussion Section: more discussion regarding WHY music might have contributed to results. There is a strong literature base regarding the therapeutic use of music to reduce anxiety or depression. Including a bit more discussion on why music may have resulted in reduced caregiver burden would strengthen conclusions.
Response 4: A paragraph has been included in the Discussion section according to the suggestion.
Point 5: A few minor editing items: Line 40: no comma needed after “death”.
Response 5: Corrected after suggestion.
Point 6: Line 44: no comma needed after “problems”.
Response 6: Corrected after suggestion.
Point 7: Line 66-68: worded awkwardly
Response 7: The sentence has been rewritten as follows “Thus, its use might alleviate burden by enhancing the caregiver’s emotional health”.

Round 2
Reviewer 1 Report
Dear authors,
thank you for addressing my points. I think that some changes are still necessary.
Abstract
Please mention that the effect is short-term. You should also highlight the fact later in the Discussion that the intervention was very brief in relation to the entire duration of the treatment. You could speak about "Short-term effects of self-chosen music..." to guide the reader in the right direction.
1. I still disagree with you that the presentation of the Methods section is appropriate. I am even more convinced that you need to revise the section thoroughly. You do have participants and the reader has the right to identify the demographic information about the participants in the study under an appropriate heading. You also do not clearly present the dependent and independent measures separately.
It is not so difficult to separate the information about who participated, what measures were applied and how they were applied. In your writing, it is all mixed up and readers have hard times to identify the relevant information.
In sum, please do not try to reinvent the wheel. Please use a strategy like everybody else. There are still bad examples in the published literature, but those are inappropriate as guidance.
2. The presentation of results as bullet points is not standard at all, and speaks to inexperienced authors. Please use the main text to present findings, OR, reconsider a Table 2 with the descriptive data. Please do not try to invent your own style of presentation. It only leads to less respect of your hard work in the community.
l.146: This para reveals the chaotic way of description you have chosen. The authors try to establish the dependent measure within the description of the procedure. That does not add up, but only confuses the reader. Please describe measures separately and give example items. Consider APA for help, not your personal opinion. Thank you.
l.160: you forgot to mention the dependent measure in the new sentences. The reader does not know at this point, what this is all about.
To be very clear: Your data will never appear in a meta analysis, if you do not provide descriptive data with means and SD of your independent and dependent measures. To avoid that and jump more or less directly to an inferential statistic seems careless to me, despite the fact that there are lots of examples like that in the literature. - You want to be more frequently cited? Construct a Table 2 without hesitation, please.
3. Please use a heading "Limitations" before. You have no argument to convince me otherwise. "Although.... (l.223), better omit this line. It does not make any sense as an intro to limitations. Do not self-applaud, no need for that. Start with:
"4. Limitations (heading)
Firstly , the participants ...
5. Conclusions
..."
This is standard to include a heading "Limitations" - not an inconvenience.
You did not get my point about the Figure. Of course, it is redundant, but it will lead to greater attention. Readers have a very brief attention-span. Within that short span, they are unlikely to read the Results section, but they are likely to catch the Figure. It is a strategy to promote your work after all.
You also did not get my point about the Limitations. A do-nothing-comparision is subject to Hawthorne-effects. This means that the people receiving attention by asking them to select music may respond better because of their greater involvement and more interactions with the practitioners. For the least, you have to argue against a Hawthorne effect by showing that the do-nothing controls received as much attention as the target group. Also, what aspect of the music helped is unclear. Why would you expect that genre or duration of listening made any difference? That is pure speculation, not a limitation.
Conclusions
"The 7-day intervention using 30-min pre-recorded music, chosen personally by the caregiver, alleviates caregivers’ burden. This intervention is simple to perform at home 243 and has no negative consequences. Therefore, its application in caregivers of palliative cancer patients is encouraged."
Better: "We demonstrate that daily listening to self-selected music for thirty minutes over a period one week can significantly enhance wellbeing of caregivers by reducing their subjective burden. The study underscores the need for complementary, cost-effective, and accessible strategies to alleviate caregiver burden." .. something like that, more to the point.
Author Response
Response to Reviewer 1 – round 2 Comments
Point 1: Dear authors, thank you for addressing my points. I think that some changes are still necessary. Abstract. Please mention that the effect is short-term. You should also highlight the fact later in the Discussion that the intervention was very brief in relation to the entire duration of the treatment. You could speak about "Short-term effects of self-chosen music..." to guide the reader in the right direction.
Response 1: We have rewritten this aspect according to the reviewers’ suggestion. Thank you very much.
Point 2: I still disagree with you that the presentation of the Methods section is appropriate. I am even more convinced that you need to revise the section thoroughly. You do have participants and the reader has the right to identify the demographic information about the participants in the study under an appropriate heading. You also do not clearly present the dependent and independent measures separately. It is not so difficult to separate the information about who participated, what measures were applied and how they were applied. In your writing, it is all mixed up and readers have hard times to identify the relevant information. In sum, please do not try to reinvent the wheel. Please use a strategy like everybody else. There are still bad examples in the published literature, but those are inappropriate as guidance.
Response 2: We have rewritten the methods according to the first revision and also following the APA guidelines as suggested, so this information is now easily identified by the reader. Thanks.
Point 3: The presentation of results as bullet points is not standard at all, and speaks to inexperienced authors. Please use the main text to present findings, OR, reconsider a Table 2 with the descriptive data. Please do not try to invent your own style of presentation. It only leads to less respect of your hard work in the community.
Response 3: The table has been included after suggestion. As the Reviewer properly indicated, the APA recommends to visually present the results in Tables or Figures; therefore, we have improved our results presentation thanks to the reviewer comments by including the Table 3 and we are grateful.
Point 4: l.146: This para reveals the chaotic way of description you have chosen. The authors try to establish the dependent measure within the description of the procedure. That does not add up, but only confuses the reader. Please describe measures separately and give example items. Consider APA for help, not your personal opinion. Thank you.
Response 4: We have modified the Method section according to the reviewer suggestion and considering APA. Subheadings have been included.
Point 5: l.160: you forgot to mention the dependent measure in the new sentences. The reader does not know at this point, what this is all about.
Response 5: It has been included as follows: “The CSI differences were analysed by applying a 2x2 mixed factorial analysis of variance…”. Thank you.
Point 6: To be very clear: Your data will never appear in a meta analysis, if you do not provide descriptive data with means and SD of your independent and dependent measures. To avoid that and jump more or less directly to an inferential statistic seems careless to me, despite the fact that there are lots of examples like that in the literature. - You want to be more frequently cited? Construct a Table 2 without hesitation, please.
Response 6: We agree. We have included the required Table (it is the Table 3 since we have kept Tables 1 and 2). Thank you very much for your detailed revision and your concern regarding our visibility.
Point 7: 3. Please use a heading "Limitations" before. You have no argument to convince me otherwise. "Although.... (l.223), better omit this line. It does not make any sense as an intro to limitations. Do not self-applaud, no need for that. Start with:
"4. Limitations (heading)
Firstly , the participants ...
- Conclusions ..."
This is standard to include a heading "Limitations" - not an inconvenience.
Response 7: We have included the Limitation heading despite the Research Manuscript Sections on Instructions for Authors (https://www.mdpi.com/journal/ijerph/instructions) and the Word Template do not include Limitations as a heading. Maybe the Journal should modify it avoiding further issues. Moreover, the introductory sentence has been modified after suggestion.
Point 8: You did not get my point about the Figure. Of course, it is redundant, but it will lead to greater attention. Readers have a very brief attention-span. Within that short span, they are unlikely to read the Results section, but they are likely to catch the Figure. It is a strategy to promote your work after all.
Response 8: To duplicate information (table and figures) does not meet APA or others guidelines, so I beg the reviewer to reconsider this point avoiding personal preferences. We have removed Mean and SD from the text, but we have included the Table 3 in the hope that it is now clearer presented and more easily identified by the reader.
According to APA: “It may be useful to include a table or figure to represent your results visually (it has been done according to the reviewer suggestion, thank you). Be sure to refer to these in your paper (done). Remember that you may present a set of findings either as a table or as a figure, but not as both (only Table). Make sure that your text is not redundant with your tables/figures (done). For instance, if you present a table of means and standard deviations, you do not need to also report these in the text (done, it has been removed). However, if you use a figure to represent your results, you may wish to report means and standard deviations in the text, as these may not always be precisely ascertained by examining the figure (accordingly, we have included the Table). Do describe the trends shown in the figure”.
We agree that Methods and Results have been improved thanks to the reviewer’s comments, we are happy with the current version of the manuscript since it has improved its quality compared to the first draft. We hope that the reviewer now finds both sections clearely presented.
Point 9: You also did not get my point about the Limitations. A do-nothing-comparision is subject to Hawthorne-effects. This means that the people receiving attention by asking them to select music may respond better because of their greater involvement and more interactions with the practitioners. For the least, you have to argue against a Hawthorne effect by showing that the do-nothing controls received as much attention as the target group. Also, what aspect of the music helped is unclear. Why would you expect that genre or duration of listening made any difference? That is pure speculation, not a limitation.
Response 9: We apologize for our missunderstood in the first revision. We have included this as a limitation, we have now understand this point. The intervention group received one extra question compared to the control group, but it may have an impact on the results. This has been detailed. Additionally, we have removed the speculation regarding genre, etc. Thank you.
Point 10: Conclusions "The 7-day intervention using 30-min pre-recorded music, chosen personally by the caregiver, alleviates caregivers’ burden. This intervention is simple to perform at home 243 and has no negative consequences. Therefore, its application in caregivers of palliative cancer patients is encouraged." Better: "We demonstrate that daily listening to self-selected music for thirty minutes over a period one week can significantly enhance wellbeing of caregivers by reducing their subjective burden. The study underscores the need for complementary, cost-effective, and accessible strategies to alleviate caregiver burden." .. something like that, more to the point.
Response 10: Thanks. We like these sentences. We have rewritten our Conclusions.

Reviewer 2 Report
Thank you for providing comments to my concerns and revising your manuscript. Unfortunately, I still have a few more questions.
Response 4: The manuscript have been reviewed by the translator of our University (University of Málaga).
My response: Are the authors saying that this review has already been carried out? Or will this still take place? I still see many typos, grammatic mistakes and not-so-smooth sentences.
Response 6: It has been rewritten as follows “another study did find significant improvements on anxiety and depression levels after a music intervention in cancer patient caregivers”.
My response: Page 2, line 78-88, the justification still isn’t convincing. Please have another go at the justification, and study objects.
Response 10: In our opinion, if we ask for preferred music but we applied the educational program, this group would know that they are not in the intervention group. All they have an intervention in which they must listen to a record session of similar duration and frequency with the same device.
My response: I understand what you did, but this isn’t a double-blinded study. It’s obvious for the control group to know that they’re not receiving the intervention. I think the authors should not call it ‘double-blinded’.
Point 14: Discussion: Line 176-186 – I suggest the authors reflect on the findings with more caution, as there were more caregivers with higher Caregiver Strain Index scores (>7) in the intervention group (61%) then in the control (53.7%) at the baseline.
Response 14: We agree and we have improved the statistic analyses (also according to another reviewer, the Reviewer 1). The mentioned differences between groups at baseline were not statistically significant. Notwithstanding, we have included this statement in limitations.
My response: I’m afraid that the discussion needs to be re-written. The authors have not discussed their results with more caution. Line 195-198 do not make sense. “This finding is novel, although an earlier clinical trial did report that listening to pre-recorded music alleviated anxiety symptoms and depression [25], conditions that are associated with burden [9,11].”
A new point: Line 238-240 “We believe than the program applied in this study is an appropriate…” this doesn’t seem to suit here in discussion. Also, “we believe..” isn’t really appropriate in the reporting RCT results. Have the authors gathered participants’ feedback? This sort of statement should come from the participants, not the authors.
Author Response
Response to Reviewer 2 – round 2 Comments
Point 1: Thank you for providing comments to my concerns and revising your manuscript. Unfortunately, I still have a few more questions. Res4: The manuscript have been reviewed by the translator of our University (University of Málaga). My response: Are the authors saying that this review has already been carried out? Or will this still take place? I still see many typos, grammatic mistakes and not-so-smooth sentences.
Response 1: We have attached the certificate of the English revision. The current submitted version has been revised. Thank you very much. It was from the University of Granada (apologize the mistake).
Point 2: Regarding Res6: It has been rewritten as follows “another study did find significant improvements on anxiety and depression levels after a music intervention in cancer patient caregivers”. My response: Page 2, line 78-88, the justification still isn’t convincing. Please have another go at the justification, and study objects.
Response 2: According to the suggestion, we have rewritten this part of the introduction.
Point 3: Regarding Res10: In our opinion, if we ask for preferred music but we applied the educational program, this group would know that they are not in the intervention group. All they have an intervention in which they must listen to a record session of similar duration and frequency with the same device. My response: I understand what you did, but this isn’t a double-blinded study. It’s obvious for the control group to know that they’re not receiving the intervention. I think the authors should not call it ‘double-blinded’.
Response 3: The project obtained competitive funding on this basic (double-blinded study), it has been also registered specifying it. In our opinion, although some limitations could be addressed, we must specified this design. It is difficult to include placebo conditions with music interventions; however, our intervention and masking have been detailed in the Methods, also, we have tried to include this concern in Limitations according to the reviewer’s comment.
Point 4: Resgarding Point14: Discussion: Line 176-186 – I suggest the authors reflect on the findings with more caution, as there were more caregivers with higher Caregiver Strain Index scores (>7) in the intervention group (61%) then in the control (53.7%) at the baseline. Res14: We agree and we have improved the statistic analyses (also according to another reviewer, the Reviewer 1). The mentioned differences between groups at baseline were not statistically significant. Notwithstanding, we have included this statement in limitations. My response: I’m afraid that the discussion needs to be re-written. The authors have not discussed their results with more caution. Line 195-198 do not make sense. “This finding is novel, although an earlier clinical trial did report that listening to pre-recorded music alleviated anxiety symptoms and depression [25], conditions that are associated with burden [9,11].”
Response 4: Our participants showed similar sociodemographic characteristics and CSI levels at baseline (p > 0.05 for all outcomes). We have tried to better present our findings by adding some sentences which add caution. The discussion has been rewritten, also the mentioned, in order to improve our manuscript before its publication.
Point 5: Line 238-240 “We believe than the program applied in this study is an appropriate…” this doesn’t seem to suit here in discussion. Also, “we believe..” isn’t really appropriate in the reporting RCT results. Have the authors gathered participants’ feedback? This sort of statement should come from the participants, not the authors.
Response 5: This part has been rewritten by removing that sentence. Thank you for your detailed revision.
